# Nanoscale nonreciprocity via photon-spin-polarized stimulated Raman scattering

Mark Lawrence[1] & Jennifer A. Dionne[1]

Time reversal symmetry stands as a fundamental restriction on the vast majority of optical systems and devices. The reciprocal nature of Maxwell's equations in linear, time-invariant media adds complexity and scale to photonic diodes, isolators, circulators and also sets fundamental efficiency limits on optical energy conversion. Though many theoretical proposals and low frequency demonstrations of nonreciprocity exist, Faraday rotation remains the only known nonreciprocal mechanism that persists down to the atomic scale. Here, we present photon-spin-polarized stimulated Raman scattering as a new nonreciprocal optical phenomenon which has, in principle, no lower size limit. Exploiting this process, we numerically demonstrate nanoscale nonreciprocal transmission of free-space beams at near-infrared frequencies with a 250 nm thick silicon metasurface as well as a fully-subwavelength plasmonic gap nanoantenna. In revealing all-optical spin-splitting, our results provide a foundation for compact nonreciprocal communication and computing technologies, from nanoscale optical isolators and full-duplex nanoantennas to topologically-protected networks.

---

[1] Department of Materials Science and Engineering, Stanford University, Stanford, CA 94305, USA. Correspondence and requests for materials should be addressed to M.L. (email: markl89@stanford.edu) or to J.A.D. (email: jdionne@stanford.edu)

n 1931, Lars Onsager established that transport properties within materials and their corresponding gradient forces must be reciprocally related, as resulting from time reversibility of the microscopic equations of motion[1]. Originally formulated for thermodynamics, Onsager's analysis also finds an important consequence in optics. Without breaking time-reversal symmetry, the permittivity and permeability tensors of any material must be symmetric[2]. It is straightforward to show that this symmetry is sufficient to ensure reciprocal light scattering in time-invariant, linear systems[3]. However, breaking of time-reversal symmetry is crucial for the protection of fragile optical components, such as laser cavities, as well as for lifting fundamental limits in photovoltaics, light storage, and sensing[4–6].

Time-reversal symmetry can be broken with a DC magnetic field, which energetically separates electronic levels possessing opposite angular momentum[7,8]. Gyromagnetic effects are reasonably strong at THz frequencies and below[9,10], but they are exceedingly weak for optical frequency light[11]. Consequently, optical isolators, i.e. one-way filters, generally have dimensions on the centimeter scale. Further miniaturization is challenged by their reliance on lossy garnet materials, which require strong magnetic biasing fields and are rarely compatible with traditional nanofabrication processes[11–13].

To achieve optical isolation at the millimeter-to-micron-scale, considerable progress has been made in creating traveling-wave refractive index distributions driven by acoustic waves, optically excited mechanical vibrations, optically excited acoustic phonons, spatiotemporally varying electronic gating, and parametric wave mixing[14–28]. Analogous to Faraday rotation, these effects provide a fixed amount of linear or orbital angular momentum to the optical system; consequently, photon absorption or amplification becomes tied to a particular incident direction, breaking reciprocity. However, because of the need for a well-defined extrinsic momentum exchange, these mechanisms cannot be employed for subwavelength nonreciprocity. In generating asymmetric scattering matrices from asymmetric electric nearfields, the nonlinear Kerr effect has also been shown to break reciprocity over subwavelength paths[29,30]. But, such schemes only work for high powered signals and require pulsed excitation because of dynamic reciprocity[31].

All-optical stimulated Raman scattering (SRS) can, in principle, break time reversal symmetry at the atomic level. In this process, an optical phonon is produced at the center of the Brillouin zone and no linear momentum is transferred to or from the photons. Instead, as pointed out by Krauss, a Faraday-like antisymmetric third order Raman susceptibility tensor emerges provided the optical pump field possesses $\pi/2$-dephased components along orthogonal crystallographic axes[32]. Using SRS, nonreciprocal amplification has been observed in optically-biased silicon waveguides[33,34]. However, due to strong anisotropy and weak amplification, these waveguide structures have still needed to be many tens to hundreds of microns long in order to build up appreciable differences in forward versus backward light transmission.

In this paper, we show how SRS can enable nanoscale optical isolation by using spin selective photon-phonon interactions. In particular, we show how Raman amplification with circularly polarized light is, rather than simply asymmetric, forbidden for one incident direction—a consequence of an angular momentum conserving spin selection rule. Then, exploiting the local nature of chiral SRS, we design nanoscale antennas which both dramatically enhance and maintain the circulation of near-infrared chiral light, culminating in nanoscale Raman-based isolation. We benchmark this process with two deeply subwavelength structures: nonreciprocal Si metasurfaces and diamond-loaded plasmonic nanoantennas. Importantly, no special symmetries are required of the mediating crystal since a spinning photon can be imprinted on almost any Raman active phonon; consequently, a host of traditional and emerging materials are available for constructing photon-spin-polarized SRS-based nonreciprocal devices.

## Results

**Nonreciprocal spin selection rules for SRS.** As illustrated in Fig. 1a, Raman scattering involves the change in frequency of an incoming pump photon following an inelastic collision with a polarizable object, resulting in the generation of a phonon. If a second signal is incident at the Stokes frequency alongside the pump, the Raman process can be stimulated, coherently amplifying the Stokes field just like stimulated emission from an excited electronic state. By exploiting this phenomenon, efficient and wavelength-tunable solid-state lasers and amplifiers have been developed[35]. The phonon energy is dependent on the object's atomic structure and represents the only intrinsic resonance of the system. In the absence of any other loss mechanisms, the difference between the pump photon energy and that of the outgoing, or Stokes, photon must therefore exactly match the phonon energy. This parametric relationship can be described as a third order correction to the electric polarization field oscillating at the Stokes frequency $\omega_s$,

$$P_i^{(3)} = \varepsilon_0 \chi_{ijkl}^{(3)} p_j p_k^* E_l \qquad (1)$$

where **p** represents the electric field at the pump frequency $\omega_p$; **E** is the electric field at the Stokes frequency $\omega_s$; $\chi_{ijkl}^{(3)}$ is the third-order susceptibility tensor of the active material with subscripts

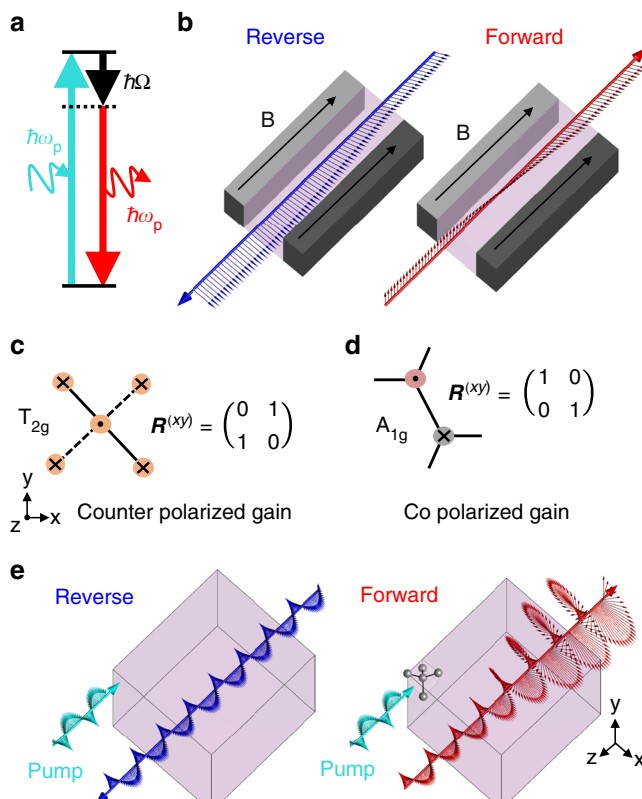

**Fig. 1** Nonreciprocal stimulated Raman scattering. **a** Energy diagram for Raman decay of a pump photon into a Stokes photon and a phonon. **b** Nonreciprocal polarization rotation due to Faraday effect in magnetized medium. **c**, **d** Phonons that produce spin-polarized nonreciprocal SRS for CPL propagating in z. Mulliken symbols and x–y Raman tensors are provided for **c** silicon, and **d** blue phosphorene. **e** One-way amplification due to stimulated Raman scattering in silicon pumped with circularly polarized light

representing vector components along the crystallographic axes $x$, $y$, and $z$; and $\varepsilon_0$ is the permittivity of free-space. Although an equivalent term that oscillates at $\omega_p$ exists, we set this term to zero throughout the rest of the paper by applying the undepleted pump approximation, which is true in the limit $|\mathbf{p}| \gg |\mathbf{E}|$. Physically this approximation is equivalent to ignoring the negligible absorption of the pump wave by the signal wave. From Eq. (1) we can see that unlike conventional gain mechanisms, Raman amplification is sensitive not only to the intensity of the pump light but also the relative phase and amplitude of the various components of the pump and Stokes electric fields. This equation forms the foundation for our study in that it represents the polarization selection rules for Raman gain.

We know from Lorentz that broken reciprocity is encoded in the difference between the off diagonal elements of the SRS susceptibility tensor $\chi_{il} = \chi_{ijkl}^{(3)} p_j p_k^*$, i.e. $|\chi_{il} - \chi_{li}| \neq 0$[3]. In the Faraday effect, for example, $\chi_{xy} = \chi_{yx}^* \propto i$, leading to unidirectional rotation of linearly polarized light, as illustrated in Fig. 1b. If we consider the pump field as a fixed optical bias and the Stokes wave to be a variable optical signal, we seek to address whether a bias field could be chosen to selectively address a specific signal mode. In particular, we are interested in amplifying just one mode of a time-reversed pair—behavior that is normally forbidden in a linear, time-invariant system. Restricting our attention to first-order Raman scattering in a perfectly crystalline structure, the third-order susceptibility contains a separate term for each zone center phonon given by

$$\chi_{ijkl}^{(3)} = \chi_{res} L R_{ij} R_{kl} \tag{2a}$$

$$L(\delta = \omega_p - \omega_s) = \frac{2\Omega\Gamma}{\Omega^2 - \delta^2 + i2\delta\Gamma} \tag{2b}$$

where $\Omega$ and $\Gamma$ are the phonon frequency and halfwidth, respectively. $\chi_{res}$ is a real number used to represent the peak Raman susceptibility which is related to the peak Raman gain. $\mathbf{R}$ is known as the Raman tensor, defined via the spontaneous Raman scattering intensity $S_{ij} \propto |\hat{E}_i \cdot R_{ij} \cdot \hat{p}_j|^2$. As we are concerned with near infrared light, $\Omega \ll \omega_p, \omega_s$ and accordingly, $\mathbf{R}$ must be symmetric[36].

To simplify our analysis of SRS selection rules we consider plane waves traveling along the $z$ cartesian axis through a medium with an isotropic linear susceptibility. As such, longitudinal electric fields can be ignored leaving the 2 by 2 Raman matrix

$$\mathbf{R}^{(xy)} = \begin{pmatrix} a & b \\ b & c \end{pmatrix}. \tag{3}$$

When operating in the coordinate frame of the crystal lattice, elements of $\mathbf{R}$ often vanish as the phonon and the relevant pump-Stokes polarization combination fail to share an irreducible representation. In the general case, however, Eqs. (1)–(3) can be used to define the pump-induced effective susceptibility tensor

$$\chi_{eff} = \chi_{res} L \left\{ \begin{pmatrix} a^2 & ab \\ ab & b^2 \end{pmatrix} |p_x|^2 + \begin{pmatrix} b^2 & cb \\ cb & c^2 \end{pmatrix} |p_y|^2 \right.$$
$$\left. + \text{Re}(p_x p_y^*) \begin{pmatrix} 2ab & ac+b^2 \\ ac+b^2 & 2cb \end{pmatrix} + (ac-b^2)\text{Im}(p_x p_y^*) \begin{pmatrix} 0 & i \\ -i & 0 \end{pmatrix} \right\}. \tag{4}$$

Nonreciprocal light flow in a uniform medium is dependent on the difference between the off-diagonal elements of the susceptibility, $|\chi_{yx} - \chi_{xy}|$. For SRS, we can see that $|\chi_{yx} - \chi_{xy}| \propto (ac - b^2)\text{Im}(p_x p_y^*)$, and so nonreciprocity will result if the pump polarization possesses some ellipticity, $\text{Im}(p_x p_y^*) \neq 0$,

and, $ac \neq b^2$. By a 45° coordinate rotation, $ac = b^2$ can be seen to represent a phonon, which couples exclusively to electric fields along a single cartesian axis; this condition is only possible for extremely anisotropic crystals. Therefore, beyond previous observations of time-reversal symmetry breaking in silicon waveguides[32,37], we expect Raman-based nonreciprocity to be ubiquitous in a range of classical and quantum photonic materials and devices.

Equation 4 also reveals an interesting special case which will be the focus of the rest of this paper. If $a = c$, $b = 0$ and the pump is circularly polarized with handedness $p_x/p_y = \mp i$, or $a = c = 0, b \neq 0$ and the pump is circularly polarized with handedness $p_x/p_y = \pm i$, an SRS susceptibility tensor proportional to $(1, \mp i; \pm i, 1)$ is produced. Examples of specific phonons that support these kinds of Raman scattering are shown in Fig. 1c, d. Fig. 1c shows the $z$ polarized phonon in silicon which possesses $S_4$ symmetry about $z$ and so mixes $x$ and $y$ polarized photons. Fig. 1d shows the $z$ polarized phonon of blue phosphorene[38] which exhibits an isotropic response and thus a diagonal Raman tensor. The key feature of the phonons depicted in Fig. 1c, d is that they couple equally to $x$ and $y$ polarized components of the incident electric field. The rotation, or spin, of the pump electric field can therefore be imprinted on to the host crystal. Many other materials support spin selective Raman tensors, including, Diamond, GaAs, GaP, barium nitrate, and potassium gadolinium tungstate. Just like the Faraday effect, the tensor $(1, \mp i; \pm i, 1)$ supports non-degenerate circularly polarized eigenstates. Uniquely, however, the susceptibility eigenvalue for one handedness, $(E_x, E_y) = (1, \mp i)$, vanishes. On resonance, or $\omega_p - \omega_s = \Omega$, $L = -i$ and so the opposite handedness, $(E_x, E_y) = (1, \pm i)$, experiences gain proportional to the pump intensity. To be clear, these states define the absolute rotation of the local electric field, which we will refer to as spin, and not the wave helicity. Counter-propagating circularly polarized plane waves of a fixed helicity, which transform into one another under time reversal, possess counter-rotating electric fields and so this form of SRS gives rise to one-way amplification, as illustrated in Fig. 1e. Importantly, the directionality of this process results purely from the modified effective material properties which appear from the point of view of the Stokes wave as a continuously rotating phonon being driven by the rotating electric pump field, which holds right down to the atomic scale.

**Enhanced circular polarization in silicon metasurfaces.** Unfortunately, $\chi_{res}$ is typically on the order of $10^{-19}(\text{mV}^{-1})^2$ and so the vast distance a Stokes signal would need to travel inside a bulk crystal, pumped with a reasonable control power, to build up a measurable amplification overshadows the local nature of SRS. However, inspired by developments in the field of dielectric metasurfaces and plasmonic nanoantennas, we show that structural resonances can be employed to shrink this distance well below the Stokes wavelength and in principle even below tens of nanometers. In particular, the metasurfaces boost the efficiency of SRS in the forward direction, suppresses transmission in the reverse direction, and maintain suppression of Raman gain in the reverse direction.

For a concrete demonstration we choose silicon which has a rather strong Raman transition, $\chi_{res} = 11.2 \times 10^{-18}(\text{mV}^{-1})^2$, $\Omega = 15.6 \, \text{THz}$, and $\Gamma \approx 53 \, \text{GHz}$, as well as a symmetric Raman tensor with zeroes on the diagonal when working in the coordinate frame of its diamond cubic lattice[39]. In the near infrared, silicon also has a high refractive index of 3.45 and a negligible linear absorption coefficient. A very popular approach for enhancing nonlinear optical phenomena has been to build ring, toroidal, or whispering-gallery resonators[40,41] out of

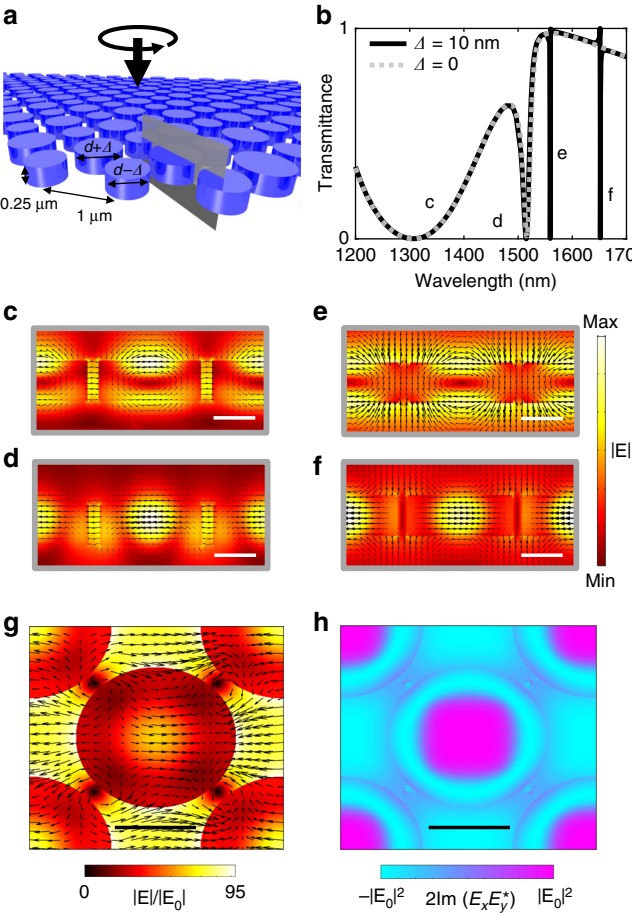

**Fig. 2** Simulating linear behavior of high $Q$ silicon metasurface. **a** Schematic of silicon metasurface, $d = 640$ nm. **b** Circularly polarized transmission through metasurface with $\Delta = 0$ (dashed gray curve) $\Delta = 10$ nm (solid black curve). **c–f** Electric field distributions, with colourmaps giving amplitude and vector directions given by arrows, in the $x = y$ plane, shown in gray in **a**, for the four resonant modes in **b**. From the electric field vector maps, it can be seen that **c** and **d** are symmetric magnetic and electric dipoles, respectively, while **e** and **f** are anti-symmetric magnetic and electric dipoles, respectively. Scale bars are 250 nm. g On resonant electric field map at the $z = 0$ plane for mode **f**. Scale bar is 325 nm long. **h** E-field spin corresponding to the map in **g**. Scale bar is 325 nm long

low-loss dielectric materials, with very weak coupling to a bus waveguide resulting in a high-quality factor (high $Q$) and a substantial electric field enhancement. But these structures are necessarily much bigger than the operational wavelength and rarely support modes with circularly polarized electric fields. On the other hand, Mie resonances in nanoscale silicon disks mimic the coupling of free space light waves to molecular orbitals. A periodic array of these so-called 'meta-atoms' is often referred to as a 'meta-surface'. As well as being designable on a subwavelength scale, it has been shown by many groups that these resonators can be easily tailored to interact with any desired polarization state[42–47].

A schematic of our Si meta-surface is shown in Fig. 2a. Each unit cell of the array consists of two 250 nm tall disks. Diffraction is forbidden for a normally incident plane wave with a wavelength greater than the array period of 1000 nm and so in the far-field the scattering simply exhibits dispersive transmission and reflection. The calculated transmittance of a circularly polarized plane wave through this metasurface is represented by the dashed curve in Fig. 2b; here, all disks are taken to have a diameter of 640 nm. The

two broad dips occurring at 1300 and 1515 nm wavelengths correspond to the usual magnetic and electric dipole modes, respectively. The resonant nature of these modes has been exploited for enhancing nonlinear frequency conversion[48–50]. But for the observation of SRS, much larger intensity enhancement is required. The solid curve in Fig. 2b shows the transmittance through a metasurface with each neighboring disks having slightly different diameters of 630 and 650 nm. In this case, two extremely sharp features appear at longer wavelengths. These features also have a dipole-like character. In fact, the electric field distribution within each disk for the 1300 nm mode and the 1559 nm mode is almost identical (see Fig. 2c, e). Likewise, the 1515 and 1659 nm modes have very similar field patterns within the silicon (see Fig. 2d, f). The very narrow line widths of the 1559 and 1659 nm modes arise due to coupling between different disks. It can be seen in Fig. 2f that neighboring dipoles are oscillating out of phase. Due to their subwavelength separation, this antisymmetry suppresses the interaction with free-space radiation, extending the resonant lifetime and leading to a large, ~80× enhancement of the incident electric field, as shown in Fig. 2g. While it may seem from Fig. 2b that the antisymmetric dipole modes cease to exist for $\Delta = 0$, this is not the case. These modes are still eigenstates of the system but are now completely bound and so do not show up in the scattering spectra of free-space waves. Therefore, the $Q$ factor, $\lambda_0$/FWHM, can in principle be increased indefinitely by asymptotically reducing the difference in size of the two disks to zero. In practice however, fabrication imperfections and intrinsic absorption provide an upper bound. As a proof of principle, we have chosen $Q$~6100. This value is fairly moderate considering $Q$'s as high as 80,000 have been observed in similar systems[51–54].

Crucially, Fig. 2g reveals that, in addition to amplifying the incident wave, the high $Q$ trapped mode concentrates the electric field close to the center of each disk. These points within the metasurface also represent axes of four-fold rotational symmetry. This symmetry ensures the electromagnetic hotspots preserve the incident polarization, as can be seen in Fig. 2h which plots the resonant field circulation under CPL illumination normalized to the field intensity. The silicon metasurface essentially acts as a light funnel for CPL, making it an excellent candidate for building a subwavelength spin-dependent Raman amplifier.

**A subwavelength Raman-biased all-optical metasurface diode.**
Working within the undepleted pump approximation, $|\mathbf{p}| \gg |\mathbf{E}|$, SRS can be modeled as a two-step process. Firstly, the linear behavior of the pump needs to be simulated at the blue Raman sideband of the trapped mode. As silicon has an optical phonon frequency of ~15.6 THz, the side band of the high $Q$ mode centered at 1658.83 nm occurs at $\lambda_p = 1527.02$ nm. Fortunately, from Fig. 3a we can see that this places the pump very close to the low $Q$ electric dipole mode which will ensure good pump-Stokes overlap in terms of the out of plane co-ordinate. Taking the silicon crystal axes to be aligned with the periodic lattice of the metasurface and using Eqs. (1)–(4), we express the modification to the relative permittivity tensor within the silicon disks as a function of the signal wavelength $\lambda_s$ by

$$\varepsilon_r(\mathbf{r}) = n_s^2 + \chi_{res} L(\delta) \begin{pmatrix} |p_y(\mathbf{r})|^2 + |p_z(\mathbf{r})|^2 & p_x^*(\mathbf{r})p_y(\mathbf{r}) & p_x^*(\mathbf{r})p_z(\mathbf{r}) \\ p_y^*(\mathbf{r})p_x(\mathbf{r}) & |p_x(\mathbf{r})|^2 + |p_z(\mathbf{r})|^2 & p_y^*(\mathbf{r})p_z(\mathbf{r}) \\ p_z^*(\mathbf{r})p_x(\mathbf{r}) & p_z^*(\mathbf{r})p_y(\mathbf{r}) & |p_x(\mathbf{r})|^2 + |p_y(\mathbf{r})|^2 \end{pmatrix},$$
$$(5)$$

where $\mathbf{p}(\mathbf{r})$ represents the pump vector field distribution and $\delta = 2\pi c \left( \frac{1}{\lambda_p} - \frac{1}{\lambda_s} \right)$. While we have included the $z$ component for completeness, it has a very small contribution to the overall

modal coupling. The trapped electric dipole mode transmission can then be recalculated to determine the effect of the induced Raman gain.

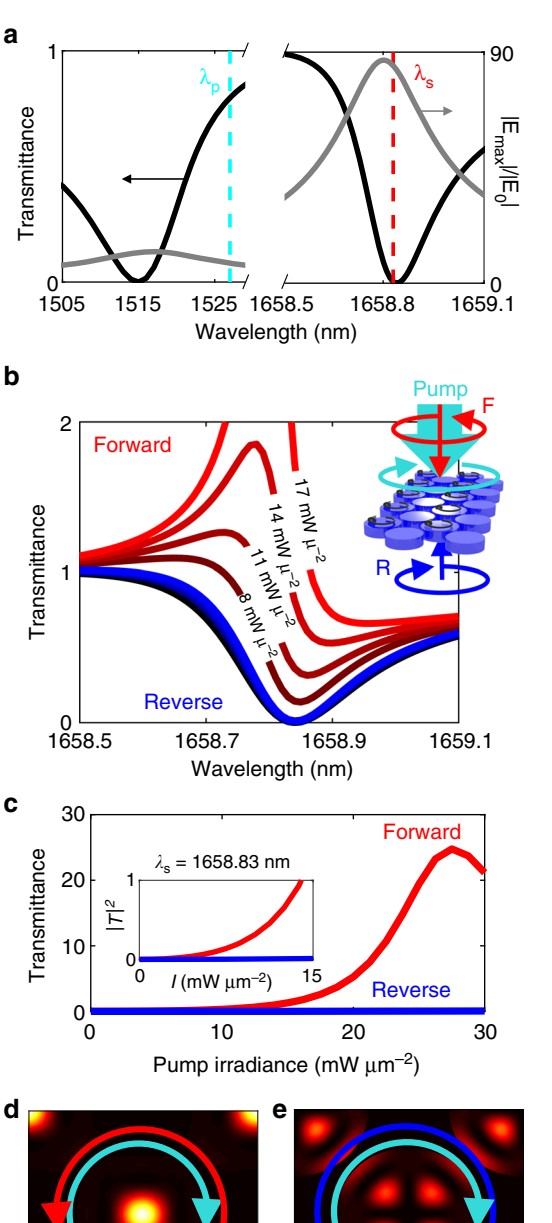

**Fig. 3** Simulating nonlinear behavior of nonreciprocal silicon metasurface. **a** Transmittance and field amplification spectra close to pump wavelength, indicated by cyan line, and Stokes wavelength, indicated by red line. **b** Evolution of transmission resonance with increasing pump power from 0.1 to 17 mW μm$^{-2}$. The pump wave is right handed circularly polarized while all signal waves are left handed circularly polarized, with red (blue) curves representing co(counter)-propagating pump and probe waves. **c** Metasurface transmittance for forward and reverse signal propagation directions as a function of pump irradiance. Inset shows zoom-in of low power behavior. **d** and **e** Spatial distributions of local contributions to normalized effective Raman susceptibility for **d** co-propagating pump and probe waves, and **e** counter-propagating pump and probe waves. Scale bars are 325 nm long

Unlike a traveling wave platform such as an optical fiber or ridge-waveguide, linear transmission is completely suppressed through our metasurface close to the Stokes wavelength. So instead of the exponential growth with propagation depicted in Fig. 1d, the red curves in Fig. 3b show that SRS manifests as a transparency window in the trapped mode spectrum. Like electromagnetic-induced absorption (EIA)[55] in an atomic vapor or optomechanically induced transparency (OMIT)[41] in a micro-resonator, this behavior can be explained by the resonant exchange between the phonons and the cavity photons. Mediated by the decay of pump photons, the efficiency of the exchange can be seen to increase with increasing pump irradiance, as shown by the red curves in Fig. 3b. This trend continues up until the Raman laser singularity, elucidated in Supplementary Note 2. As represented schematically in the inset of Fig. 3b, we probe the reciprocity of the system by fixing the handedness of the circularly polarized bias and comparing the transmission for time reversed signals, i.e. circularly polarized plane waves traveling in opposite directions. For counter propagating pump-Stokes waves, represented by the blue curves in Fig. 3b, extremely small phonon-photon coupling is seen even for the largest pump irradiance of 17 mW μm$^{-2}$.

Figure 3c compares a resonant signal passing through the metasurface in forward and reverse propagation directions for increasing pump power. As seen by the persistently low value of the blue curve in Fig. 3c, our metasurface is fully opaque under reverse illumination, but becomes transparent for forward illumination when pumping above 7 mW μm$^{-2}$, peaking at 27 mW μm$^{-2}$; higher pump powers reduce the nonreciprocal response due to the onset of lasing. Note that this result is distinct from the case illustrated in Fig. 1d for a bulk silicon crystal; there, the signal incident from one direction is amplified while from the opposite direction transmission is neither amplified nor suppressed. Our ultrathin silicon metasurface, with a thickness of just $\lambda_s/7$, therefore exhibits highly nonreciprocal transmission.

As with photonic crystal Raman lasers, the pump threshold can be further reduced by simultaneously amplifying both the pump and probe light fields with high $Q$ resonances. Supplementary Note 3 describes a structure for which the high $Q$ magnetic and electric modes of Fig. 2b, e, f are separated by exactly 15.6 THz. Here the pump irradiance required to break reciprocity drops by three orders of magnitude. Also, for simplicity SRS is the only nonlinear phenomenon included in Fig. 3. In Supplementary Note 4, we analyze the effects of two photon absorption, cross phase modulation, and cross amplitude modulation, finding that strong nonreciprocity persist with similar power thresholds.

Remarkably, the spin selectivity of bulk SRS is almost fully replicated, despite the strong spatial variation associated with both the pump and signal electric fields. For weak amplification the modal dependence of the Mie resonant structure can be understood in terms of an effective Raman susceptibility,

$$\chi_{\text{eff}} = \frac{\chi_{\text{res}}L}{N_{\text{p}}N_{\text{E}}} \int_{\text{Si}} \left( E_x^*(\mathbf{r}), E_y^*(\mathbf{r}) \right) \cdot \begin{pmatrix} |p_y(\mathbf{r})|^2 & p_x^*(\mathbf{r})p_y(\mathbf{r}) \\ p_y^*(\mathbf{r})p_x(\mathbf{r}) & |p_x(\mathbf{r})|^2 \end{pmatrix} \cdot \begin{pmatrix} E_x(\mathbf{r}) \\ E_y(\mathbf{r}) \end{pmatrix} dxdydz, \tag{6}$$

where $N_{\text{E}} = Z_0 \int_{\text{unit cell}} \text{Re}(\mathbf{E} \times \mathbf{H}^*)_z dxdy$ and $N_{\text{p}} = Z_0 \int_{\text{unit cell}} \text{Re}(\mathbf{p} \times \mathbf{H}_{\mathbf{p}}^*)_z dxdy$ are normalization coefficients. Figure 3d, e plot the integrand of Eq. (6) on resonance, $\lambda_s = 1658.83$ nm, with $\chi_{\text{res}}$ and $L$ set to one. Because of the highly localized nature of the optical hotspots, the far-field polarization is faithfully replicated in the near-field of the disks. As a result, the overlap between co-rotating and counter-rotating polarization states differ by more than an order of magnitude, yielding strong nonreciprocity. There are however clear

regions in Fig. 3e where the coupling is enhanced, which explains the small but finite SRS observed for the reverse signal illumination (blue curves of Fig. 3b). In these regions the electric dipole modes support linearly polarized electric fields with $E_x = \pm E_y$, meaning they provide coupling between the silicon phonons and photons of either handedness. This issue is discussed in more detail in Supplementary Note 5. By focusing on this metric, metasurfaces with improved isolation performance could most certainly be designed.

**A stimulated-Raman-biased plasmonic nanoantenna.** In Figs. 2 and 3 we have presented a novel platform for breaking reciprocity over subwavelength optical paths. We now show how this concept can be extended to structures with subwavelength lateral confinement as well, using plasmonic nanoantennas. Similar to dielectric resonators, metallic cavities can significantly enhance an incident light field. But, unlike dielectric systems which rely on extending the lifetime of photonic modes, so-called surface plasmon polaritons exploit free electron oscillations to confine light spatially within nanoscale volumes[56]. This phenomenon has already been applied to Raman sensors, enabling single molecule sensitivity[57,58]. In our case, not only do plasmons loosen the spectral requirements for observing nonlinear frequency conversion, but they also open the possibility for breaking reciprocity with a device that is subwavelength in all three dimensions.

Figure 4a illustrates one proposed structure, consisting of silver bow tie antennas symmetrically arranged around a diamond nanoparticle. The gaps between opposing strips of silver act as nanoscale capacitors. As such, the electric field inside each gap at the plasmon resonance varies with inverse proportionality to both the gap size and dielectric constant of the filling medium[58,59]. Selecting diamond as the Raman active medium, which has a lower refractive index than silicon, and keeping the dielectric region below 10 nm, sufficient optical intensity can be generated to observe nonlinear behavior. The rotational symmetry guarantees circular polarization is maintained at the center of the device. Unlike the silicon metasurface, however, linearly polarized plasmonic fields are not insignificant. As shown in the inset of Fig. 4b, to avoid spin insensitive Raman amplification we place a small, 4 nm diameter diamond nanoparticle at the center of the gap. While it promotes polarization selectivity, the small size of the diamond inclusion does decrease the SRS efficiency, hence increasing the pump irradiance needed. Nevertheless, pumping with 700 mW μm$^{-2}$ can be seen to open a one-way transparency window with a resonator just 40 nm thick and 300 nm wide, as seen in Fig. 4b.

## Discussion

In summary, we have presented a new all-optical, magnet-free scheme for breaking microscopic photonic time-reversal symmetry. The scheme relies on SRS in which an intense pump light wave and a weaker probe light wave mix coherently while generating lattice vibrations. Using the polarization selection rules for SRS in bulk crystals, we have shown that asymmetric-induced susceptibilities, the hallmark of nonreciprocity, should be commonplace in Raman amplifiers. We have also shown that amplification can be completely suppressed in one direction by using circularly polarized pump and signal waves traveling along a rotationally symmetric or antisymmetric axis of the mediating crystal. We significantly enhance this phenomenon with a high $Q$ silicon metasurface, numerically demonstrating near-infrared nonreciprocal light transmission through a deeply subwavelength (250 nm~$\lambda_s$/7) structure. We also employ plasmonics to design a deeply-subwavelength, 40 nm~$\lambda_s$/22, nonreciprocal nanoantenna.

Throughout our numerical investigation, we have considered left-handed CPL probe waves. Although Raman scattering breaks

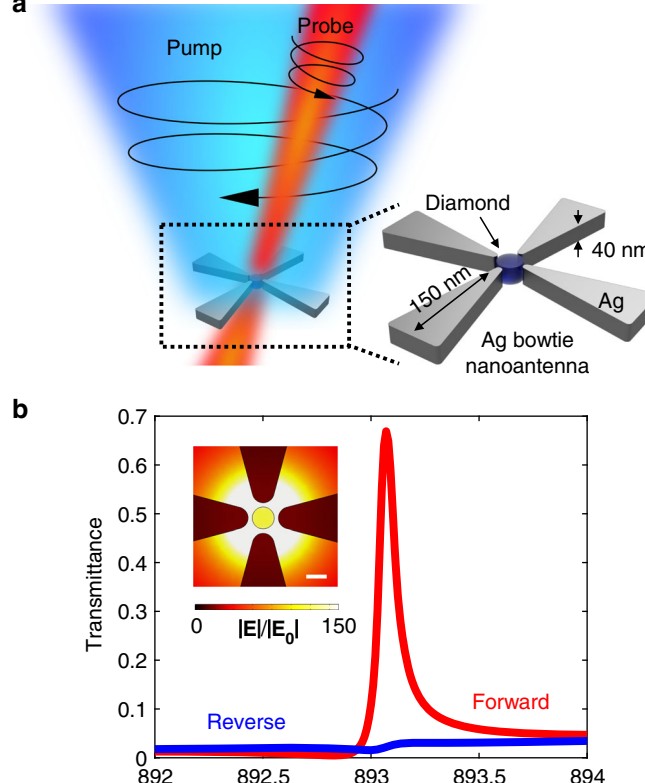

**Fig. 4** Simulating behavior of nonreciprocal plasmonic resonator. **a** Schematic of rotationally symmetric, diamond loaded bow-tie nanoantenna. **b** One-way transparency in plasmonic resonance for circularly polarized pump with intensity of 700 mW μm$^{-2}$. Inset shows field enhancement within diamond loaded gap. Surrounding medium has $n = 1$. Scale bar is 4 nm long

time-reversal symmetry it does not break spatial inversion symmetry. If we mirror the entire system except for the pump field, including both the dielectric structure and the optical signal fields, about the center plane of the metasurface/nanoantenna we find the system unchanged apart from a sign flip of the signal helicity. As the local electric field rotates in opposite directions for opposite helicities, right-handed CPL will experience the same nonreciprocal response as left-handed CPL but with the roles of forward and reverse direction exchanged, which we confirm in Supplementary Fig. 6. Just as cross polarizers are needed to turn a magneto-optic crystal into a Faraday isolator, a single handedness of CPL must be selected to fully suppress transmission through our metasurface in the reverse direction. A traditional way of building such a circular polarizer is given in Supplementary Fig. 7. From a nanophotonics perspective, thankfully, unlike time-reversal symmetry, spatial inversion symmetry can be broken by structuring standard linear dielectric materials. Many examples of chiral metamaterials and metasurfaces, using both dielectric and metallic resonators, have appeared in the literature. So, while beyond the scope of the present study, we believe that designing chiral versions of the devices shown in Figs. 2–4 that are capable of full isolation should be well within reach[60–62].

We also note that while the necessary pump irradiances are high, they are achievable; and considering the generality of spin-dependent SRS and the fact that $Q$ factors of >10$^4$ have been measured in structures like those considered here, much lower pump thresholds should be possible[51,53]. Additionally, similar dipolar mode profiles can be generated in high $Q$ L1 and H1

photonic crystal defect cavities[63,64]. Due to the reduced mode volume of these structures, the required power densities for SRS could be met with just a few milliwatts, comparable to active integrated magneto-optical isolators[65]. By moving to a doubly resonant design even lower pump powers in the tens of microwatts should be possible (see Supplementary Fig. 1). Importantly, time-reversal symmetry breaking within the proposed phonon-based platform occurs independent of the signal field strength. Beyond miniaturizing optical isolators, we anticipate time asymmetric SRS may enable a host of novel photonic and plasmonic devices, from self-isolated nanoscale lasers and nonreciprocal phase gradient metasurfaces to tunable non-Hermitian photonic crystals.

## Data availability

The data that support the plots within this paper and other findings of this study are available from the corresponding author upon request.

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

## Acknowledgements

We thank David Barton and Jefferson Dixon for their feedback on this manuscript. Funding for this work was generously provided by an AFOSR PECASE grant (FA9550-15-1-0006) as well as a NSF EFRI grant (1641109). We also gratefully acknowledge support from the Alfred P. Sloan Foundation.

## Author contributions

M.L. conceived the idea and performed the theoretical analysis and numerical simulations. J.A.D. supervised the project. M.L. and J.A.D. discussed the data, analyzed the results, and wrote the manuscript.

## Additional information

**Competing interests:** The authors declare no competing interests.

