## [Peer Review File · Nature Communications]

Reviewers' Comments:

Reviewer #1:

Remarks to the Author:

In this work the authors introduce and study theoretically and numerically a novel scheme for breaking time-reversal symmetry and attaining nonreciprocity on the nanoscale, using stimulated Raman scattering. The work is inspired by previous (well-known) ones on Raman-induced nonreciprocity in silicon waveguides (Refs. [24]-[26]), non-reciprocity on the subwavelength scale (Ref. [21]), and many works on the resonant enhancement of nonlinear and chiral effects with high-Q resonances in nanophotonics [see, e.g., Science 358, eaan5196 (2017), ACS Photonics 5, 2669 (2018), etc], but it nonetheless manages to take a novel route, and the eventual effect, as well as structures, it identifies and proposes are sufficiently original and new. The initial concise theory, which is clearly outlined, is followed by full-wave simulations of realistic structures, which corroborate and verify the attainment of robust nanoscale nonreciprocity - independent of the signal field strength.

All in all, in light of the significant recent effort on the attainment of non-magnetic nonreciprocity (ideally, at the nanoscale), and the appreciable novelty of the present work, I feel it makes an interesting and timely contribution to the wider field, which should in principle merit publication in NCOMMs after the authors address the following points:

- 1) An important figure-of-merit for many envisaged applications of non-magnetic nonreciprocal elements is their power consumption and how that compares with state-of-the-art magnetic isolators. In other words, it is undoubtedly interesting and welcome to conceive and design nanoscale non-magnetic isolators, etc, but particularly for integrated nano-photonics/-electronics applications their energy efficiency is a key parameter. To this end, it appears to me that the authors' nonlinear (high-intensity pumping) approach might be somewhat energy-costly, and for this reason I think it would be important for the authors to explicitly clarify, in a revised version of their work, the energy/power requirements of their scheme and how they compare with state-of-the-art (integrated) magnetic isolators.
- 2) On p. 9, it is shown that with the aid of dissymmetry between nanodisks, one may suppresses the interaction with free-space radiation, and extend the resonant lifetime, which finally (according to Fig. 2) leads to very sharp resonances and thus very high Q values. Then, in line 14, it is argued that this Q factor can be increased indefinitely by asymptotically reducing the difference in size of the two disks to zero. These two statements might be somewhat ambiguous to prospective readers since reducing the dissymmetry to zero results in the dashed grey-curve transmission spectra of Fig. 2 which apparently has lower Q compared with the solid black curve of the corresponding responses.
- 3) In Fig. 2a it might be better if the asymmetry value between the two disks was denoted by a different symbol rather than "delta", because the same symbol is also used on p. 11, just after Eq. (5).
- 4) In Eq. (1), for more clarity, it might be better if the authors explicitly state that: (i) $P_i^{(3)}$ is the electric polarization at the Stokes frequency (ω_s); (ii) the subscripts represents different vector components along the crystallographic axes (x; y; z); (iii) define χ_{res} ; (iv) immediately after Eq. (1), replace χ with $\chi^{(3)}$.
- 5) In Ref. [23], from where this work was inspired, it is stated that nonreciprocity results from the presence of longitudinal waveguide mode-field components, which are out of phase with respect to the transverse fields by $\pi/2$. The authors are, thus, here asked if it might indeed be possible to interpret those modes in terms of optical chirality (OC), and then use OC instead of field helicity (introduced in Fig. 2e)? The authors may, to this end, refer to the afore-cited ACS Photonics paper, which has similarly exploited a metasurface composed of silicon disks and calculated the

chiral near fields in the near-field zone of the metasurface.

6) It would be helpful if the authors could introduce a figure-of-merit for the amplification / nonreciprocity distance that would incorporate all affecting parameters (such as, pump power, pump-field helicity, optical chirality, etc) and provide key insights into the trade-offs that are to be accommodated in the design and conception of a typical nanophotonic structure for nonreciprocal SRS platforms. Then, by using that FOM, they might be able to make a better comparison between optically-biased silicon waveguides and the herein-proposed dielectric metasurface/plasmonic nanoantennas. It would be interesting to see whether such a FOM could be linked to the optical chirality of light or not? If yes, the authors could introduce structured chiral light as a new solution to nanoscale nonreciprocity.

7) Important, and of potential interest to the manuscript's broader potential audience, recent relevant works are currently missing and (together with the afore-cited works in my comments above) should be cited in a revised version of the manuscript: Nature Communications 6, Article number: 8359 (2015); Science 356, 1260 (2017); Nature Photonics 12, 613 (2018); Nature Communications 9, Article number: 3740 (2018); Nature Communications 9, Article number: 4320 (2018).

Reviewer #2:

Remarks to the Author:

This manuscript provides a computational analysis of a strategy for nanoscale non-reciprocity based on stimulated Raman scattering. The manuscript is very intriguing and appears to be of very high scientific and technical quality. Most importantly, I think the authors have justified their claims about the scalability (miniaturization) enabled by the strategy. I have some minor points and comments (listed below) that are intended to aid the clarity of future drafts of the manuscript.

However, I do have a concern about an important focus of the discussion of the manuscript, that should be addressed before I imagine the manuscript would be suitable and of interest to a broad scientific audience. Specifically, the figure of merit for the proposed structure (in figure 3c for example) is based on the difference in the transmittance of the forward and reverse signal beam. To prove the claim that the structure is indeed non-reciprocal, the authors compared situations with opposite circular polarization of the forward (right-handed) and reverse (left-handed) signal beam. While this is important to confirm that the time-reversed signal beam cannot propagate through the metasurface, this is not the full analysis I would expect to confirm the technological utility of the design for application as an optical isolator or "one-way filter". If the metasurface were used to transmit the signal beam to some receiver, but prohibit the reverse transmission of the signal beam that was not entirely collected by the receiver, the spin-state of the signal beam may not necessarily flip. For example, while specular reflection reverses circular polarization, a fiber optic tracing out a U-turn could preserve handedness. Moreover, I believe it would be reasonable to analyze the transmittance of the reverse signal beam with purely left-handed and purely right-handed circular polarization, or as a function of the helicity of the reverse beam (with fixed pump helicity), because the helicity of the reverse signal beam may not be well defined in general. Other specific comments follow:

1. The presence of the rotating phonon mode seems to be essential for realizing the SRS-based optical isolator, but it is not conveyed clearly in the text (I guess it is a special case of equation 4).
2. It is unclear if the design actually prohibits transmission of the reverse signal, or alternatively, if the design allows transmission of the reverse signal but does not provide gain for it. Perhaps this could be better emphasized if already stated.
3. It could be helpful to better define some terms or clarify meaning, especially for a more general audience:

- a. "undepleted pump approximation"
- b. "fixed rotation of the bias field phonon system"
- c. "traceless Raman tensor"

Reviewer reply

Reviewers' comments:

Reviewer #1 (Remarks to the Author):

In this work the authors introduce and study theoretically and numerically a novel scheme for breaking time-reversal symmetry and attaining nonreciprocity on the nanoscale, using stimulated Raman scattering. The work is inspired by previous (well-known) ones on Raman-induced nonreciprocity in silicon waveguides (Refs. [24]-[26]), non-reciprocity on the subwavelength scale (Ref. [21]), and many works on the resonant enhancement of nonlinear and chiral effects with high-Q resonances in nanophotonics [see, e.g., Science 358, eaan5196 (2017), ACS Photonics 5, 2669 (2018), etc], but it nonetheless manages to take a novel route, and the eventual effect, as well as structures, it identifies and proposes are sufficiently original and new. The initial concise theory, which is clearly outlined, is followed by full-wave simulations of realistic structures, which corroborate and verify the attainment of robust nanoscale nonreciprocity - independent of the signal field strength.

All in all, in light of the significant recent effort on the attainment of non-magnetic nonreciprocity (ideally, at the nanoscale), and the appreciable novelty of the present work, I feel it makes an interesting and timely contribution to the wider field, which should in principle merit publication in NCOMMs after the authors address the following points:

We thank the reviewer for concluding that we have “taken a novel route” to nonreciprocity and finding our work “original and new”.

1) An important figure-of-merit for many envisaged applications of non-magnetic nonreciprocal elements is their power consumption and how that compares with state-of-the-art magnetic isolators. In other words, it is undoubtedly interesting and welcome to conceive and design nanoscale non-magnetic isolators, etc, but particularly for integrated nano-photonics/-electronics applications their energy efficiency is a key parameter. To this end, it appears to me that the authors' nonlinear (high-intensity pumping) approach might be somewhat energy-costly, and for this reason I think it would be important for the authors to explicitly clarify, in a revised version of their work, the energy/power requirements of their scheme and how they compare with state-of-the-art (integrated) magnetic isolators.

We agree that energy consumption is an important consideration for the commercialization of nanophotonic devices. For the extended metasurface we present in figures.2-3, a defocused beam a few hundred micrometers in diameter is needed to excite the lattice resonance, which pushes the required pump powers to a few hundred watts. However, very similar mode profiles with similar, if not higher, Q factors can be achieved in photonic crystal defect cavities. By extending our scheme to these structures, pump powers of a few milliwatts should be sufficient which is comparable to electromagnet based integrated isolators. Using a doubly resonant scheme, as demonstrated in the SI, pump powers in the microwatts should even be possible. We should also point out that in active integrated isolators the consumed power goes in to heating the current carrying wires, while at the same time the optical signal is being attenuated via absorption in the magnetic material. Although high pump power densities are needed in our approach, most of that energy remains in the pump mode and so a single source could be recycled to bias multiple devices. And, much of the power that is lost from the pump goes directly to amplifying the signal. All in all, we believe our platform is very attractive from an energy perspective. To highlight this in the text we have added the following to the concluding paragraph:

“Additionally, similar dipolar mode profiles can be generated in high Q L1 and H1 photonic crystal defect cavities.^{63,64} Due to the reduced mode volume of these structures, the required power densities for SRS could be met with just a few milliwatts, comparable to active integrated magneto-optical isolators.⁶⁵ By moving to a doubly resonant design, included in the supporting information, even lower pump powers in the tens of microwatts should be possible.”

63. Thijssen, A. C. T., Cryan, M. J., Rarity, J. G. & Oulton, R. Transfer of arbitrary quantum emitter states to near-field photon superpositions in nanocavities. *Opt. Express* **20**, 22412 (2012).

64. Takagi, H. *et al.* High Q H1 photonic crystal nanocavities with efficient vertical emission. *Opt. Express* **20**, 28292 (2012).

65. Huang, D., Pintus, P. & Bowers, J. E. Towards heterogeneous integration of optical isolators and circulators with lasers on silicon [Invited]. (2018).

doi:10.1364/OME.8.002471

2) On p. 9, it is shown that with the aid of dissymmetry between nanodisks, one may suppresses the interaction with free-space radiation, and extend the resonant lifetime, which finally (according to Fig. 2) leads to very sharp resonances and thus very high Q values. Then, in line 14, it is argued that this Q factor can be increased indefinitely by asymptotically reducing the difference in size of the two disks to zero. These two statements might be somewhat ambiguous to prospective readers since reducing the dissymmetry to zero results in the dashed grey-curve transmission spectra of Fig, 2 which apparently has lower Q compared with the solid black curve of the corresponding responses.

To lift the ambiguity, we have added the following to the description of figure 2: “While it may seem from Fig. 2b that the antisymmetric dipole modes cease to exist for $\Delta=0$, this is not the case. These modes are still eigenstates of the system but are now completely bound and so do not show up in the scattering spectra of free-space waves.”

3) In Fig. 2a it might be better if the asymmetry value between the two disks was denoted by a different symbol rather than "delta", because the same symbol is also used on p. 11, just after Eq. (5).

We thank the reviewer for pointing out the double use of δ . We have changed δ in fig. 2a to Δ to avoid confusion.

4) In Eq. (1), for more clarity, it might be better if the authors explicitly state that: (i) $P_i^{(3)}$ is the electric polarization at the Stokes frequency (ω_s); (ii) the subscripts represents different vector components along the crystallographic axes (x; y; z); (iii) define χ_{res} ; (iv) immediately after Eq. (1), replace χ with $\chi^{(3)}$.

We thank the reviewer for pointing to unclear notation. The suggested changes have been made to the description of equation 1:

“This parametric relationship can be described as a third order correction to the electric polarization field oscillating at the Stokes frequency ω_s ,

$$P_i^{(3)} = \epsilon_0 \chi_{ijkl}^{(3)} p_j p_k^* E_l \quad (1)$$

where p represents the electric field at the pump frequency ω_p ; E is the electric field at the Stokes frequency ω_s ; $\chi^{(3)}$ is the third order susceptibility tensor of the active material with subscripts representing vector components along the crystallographic axes x, y , and z ; and ϵ_0 is the permittivity of free-space. “

The following line has also been added directly after equation 2b:

“ χ_{res} is a real number used to represent the peak Raman susceptibility which is related to the peak Raman gain.”

5) In Ref. [23], from where this work was inspired, it is stated that nonreciprocity results from the presence of longitudinal waveguide mode-field components, which are out of phase with respect to the transverse fields by $\pi/2$. The authors are, thus, here asked if it might indeed be possible to interpret those modes in terms of optical chirality (OC), and then use OC instead of field helicity (introduced in Fig. 2e)? The authors may, to this end, refer to the afore-cited ACS Photonics paper, which has similarly exploited a metasurface composed of silicon disks and calculated the chiral near fields in the near-field zone of the metasurface.

As seen from equation 1, SRS can be described purely in terms of the electric susceptibility. In both [23] and the present study, rotating pump electric fields (i.e. out of phase transverse electric field components) are shown to break time reversal symmetry by producing antisymmetric permittivity tensor elements. Ref [23] does indeed emphasize the importance of longitudinal field components in SRS biased waveguides. This importance is, however, more a feature of ridge waveguides than Raman scattering. The broken rotational symmetry of the waveguide breaks the degeneracy between TE and TM modes. With only 1 transverse field component left for each mode, longitudinal components are required to have rotating electric fields. While in the case of a thin waveguide longitudinal fields are accompanied by strong field gradients, to first order E-field gradients nor magnetic fields, and thus also optical chirality, play a role in SRS. Another way to see this is through symmetry. Optical chirality transforms symmetrically under time reversal but antisymmetrically under spatial inversion, which allows a chiral molecule of a given handedness to couple primarily to CPL of one handedness travelling in both directions. Instead, SRS interacts with locally rotating electric fields which transform antisymmetrically under time reversal but symmetrically under spatial inversion, allowing CPL of a given handedness to experience different gain for different propagation directions. For clarity, “helicity” has been exchanged for electric field spin in figure 2e and the main text.

6) It would be helpful if the authors could introduce a figure-of-merit for the amplification / nonreciprocity distance that would incorporate all affecting parameters (such as, pump power, pump-field helicity, optical chirality, etc) and provide key insights into the trade-offs that are to be accommodated in the design and conception of a typical nanophotonic structure for nonreciprocal SRS platforms. Then, by using that FOM, they might be able to make a better comparison between optically-biased silicon waveguides and the herein-proposed dielectric metasurface/plasmonic nanoantennas. It would be interesting to see whether

such a FOM could be linked to the optical chirality of light or not? If yes, the authors could introduce structured chiral light as a new solution to nanoscale nonreciprocity.

The most direct measure of our Raman based nonreciprocal metasurfaces is the asymmetry of their scattering parameters. However, the nearfield behavior underlying the scattering parameters can be useful for rationally designing and optimizing nanophotonic structures. The evaluation of equation 6 for different propagation directions represents the best way of constructing a nearfield figure of merit. While it is difficult to compare resonant and non-resonant devices, we have added the following section to the SI in which we apply equation 6 to better understand SRS in our dielectric metasurface and a silicon waveguide, with the reference “This issue is discussed in more detail in the supporting information.” after equation 6 in the main text.

Design principles for SRS based nonreciprocal nanophotonics

A bulk silicon crystal pumped with CPL represents the ideal case for nonreciprocal SRS. A counter-propagating signal is maximally amplified, while a co-propagating signal is entirely unaffected by the pump. A nanostructured device therefore has three roles to play: 1) to boost the efficiency of SRS in the forward direction, 2) suppress transmission in the reverse direction, 3) and maintain suppression of Raman gain in the reverse direction. Our metasurface design achieves 1 and 2 via a high Q resonance and 3 with rotational symmetry.

Two figures of merit can be defined using the linear nearfields and equation 6 in the main text which can inform the design of nonreciprocal SRS based devices. The overall SRS efficiency depends on evaluating equation 6 for the forward signal illumination configuration χ_f , while the ratio of evaluating equation 6 for forward and reverse configurations χ_f/χ_r relates to the asymmetry in the effective Raman gain coefficient. For our silicon metasurface, $\chi_f=3.62e9$ and $\chi_f/\chi_r=7.42$. In S7 we compare the central plane of our silicon metasurface, which well represents the entire mode due to its weak variation in z, and the TE mode of a [100] silicon waveguide. While the waveguide shows large asymmetry $\chi_f/\chi_r=527$, the SRS enhancement factor is significantly weaker, with $\chi_f=2.05e4$. As discussed in the main text, the non-zero overlap between co-rotating pump and signal fields, reproduced in S7a, originates from linearly polarized fields. For a waveguide, the transverse and longitudinal fields have a uniform $\pi/2$ phase difference and so the pump/signal overlap in the reverse configuration remains small despite the spatial distributions of $|E_z|$ and $|E_y|$ being very different. Comparing S7e and f, however, we see that this cancellation is detrimental to the efficiency of SRS in the forward direction as pump-signal interactions are forbidden at the center of the structure where the E-field is strongest. On the other hand, S7c and d are almost identical, showing that pump-signal interactions are near optimal in the metasurface.

Finally, we note that the waveguide only shows nonreciprocal amplification and does not suppress transmission in the reverse direction. The strong asymmetry of the waveguide also relies on the [100] crystal orientation. The field overlap in S7d, and thus reverse Raman gain, quickly grows as the crystal is rotated towards [110]. Much weaker nonreciprocity should therefore be expected upon constructing a ring cavity from such a waveguide. While χ_f and χ_f/χ_r are important factors for designing and improving the performance of nonreciprocal Raman based devices, ultimately, the suitability of such a system within a given application must be determined by the scattering parameters and from figures 3b and c we can clearly see that our subwavelength metasurface exhibits a strong nonreciprocal response.

S7: Effective Raman susceptibility for silicon metasurface (a-c) and silicon waveguide (d-f). Raman susceptibility, i.e. pump and probe overlap with Raman tensor, in reverse direction (a and d), and forward direction (b and e). Norm squared product of pump and probe fields (c and f). scale bar is 325 nm (a-c) and 220 nm (d-f).

7) Important, and of potential interest to the manuscript's broader potential audience, recent relevant works are currently missing and (together with the afore-cited works in my comments above) should be cited in a revised version of the manuscript: Nature Communications 6, Article number: 8359 (2015); Science 356, 1260 (2017); Nature Photonics 12, 613 (2018); Nature Communications 9, Article number: 3740 (2018); Nature Communications 9, Article number: 4320 (2018).

The suggested references have been added to the manuscript.

5. Tsakmakidis, K. L. *et al.* Breaking Lorentz reciprocity to overcome the time-bandwidth limit in physics and engineering. *Science* **356**, 1260–1264 (2017).
6. Lau, H.-K. & Clerk, A. A. Fundamental limits and non-reciprocal approaches in non-Hermitian quantum sensing. *Nat. Commun.* **9**, 4320 (2018).

8. Tokura, Y. & Nagaosa, N. Nonreciprocal responses from non-centrosymmetric quantum materials. *Nat. Commun.* **9**, 3740 (2018).
21. Kittlaus, E. A., Otterstrom, N. T., Kharel, P., Gertler, S. & Rakich, P. T. Non-reciprocal interband Brillouin modulation. *Nat. Photonics* **12**, 613–619 (2018).
29. Mahmoud, A. M., Davoyan, A. R. & Engheta, N. All-passive nonreciprocal metastructure. *Nat. Commun.* **6**, 8359 (2015).
47. Mohammadi, E. *et al.* Nanophotonic Platforms for Enhanced Chiral Sensing. *ACS Photonics* **5**, 2669–2675 (2018).
56. Tsakmakidis, K. L., Hess, O., Boyd, R. W. & Zhang, X. Ultraslow waves on the nanoscale. *Science* **358**, eaan5196 (2017).

Reviewer #2 (Remarks to the Author):

This manuscript provides a computational analysis of a strategy for nanoscale non-reciprocity based on stimulated Raman scattering. The manuscript is very intriguing and appears to be of very high scientific and technical quality. Most importantly, I think the authors have justified their claims about the scalability (miniaturization) enabled by the strategy. I have some minor points and comments (listed below) that are intended to aid the clarity of future drafts of the manuscript.

We thank the reviewer for stating that our work is “very intriguing and appears to be of very high scientific and technical quality”

However, I do have a concern about an important focus of the discussion of the manuscript, that should be addressed before I imagine the manuscript would be suitable and of interest to a broad scientific audience. Specifically, the figure of merit for the proposed structure (in figure 3c for example) is based on the difference in the transmittance of the forward and reverse signal beam. To prove the claim that the structure is indeed non-reciprocal, the authors compared situations with opposite circular polarization of the forward (right-handed) and reverse (left-handed) signal beam. While this is important to confirm that the time-reversed signal beam cannot propagate through the metasurface, this is not the full analysis I would expect to confirm the technological utility of the design for application as an optical isolator or “one-way filter”. If the metasurface were used to transmit the signal beam to some receiver, but prohibit the reverse transmission of the signal beam that was not entirely collected by the receiver, the spin-state of the signal beam may not necessarily flip. For example, while specular reflection reverses circular polarization, a fiber optic tracing out a U-turn could preserve handedness. Moreover, I believe it would be reasonable to analyze the transmittance of

the reverse signal beam with purely left-handed and purely right-handed circular polarization, or as a function of the helicity of the reverse beam (with fixed pump helicity), because the helicity of the reverse signal beam may not be well defined in general. Other specific comments follow:

The focus of our paper is breaking reciprocity/time reversal symmetry on the subwavelength/nano scale. The reason we have focused on this problem is that mechanisms available for breaking reciprocity are very limited, usually requiring a magnetic field. For the specific application of an isolator, the problem must be reduced to just two ports. With a free-space device, this means selecting a single polarization state to be transmitted. Distinguishing between different polarization states is relatively straightforward. In the case of a Faraday isolator, crossed linear polarisers are used. As our SRS biased metasurfaces are centrosymmetric, while all CPL inputs experience highly nonreciprocal transmission, the forward or amplifying directions for opposite handedness are related by inversion along the metasurface normal. Similar to the Faraday effect, isolation can be achieved by placing a circular polarizer in front of the metasurface. Much work on nanoscale circular polarisers, theoretical and experimental, has also been reported in the literature. While beyond the scope of the current study, by amplifying SRS within a metasurface made from chiral nanoantennas, a nanoscale isolator could be constructed. For clarity the following discussion has been added to the final paragraph:

“Throughout our numerical investigation, we have considered left handed CPL probe waves. Although Raman scattering breaks time reversal symmetry it doesn’t break spatial inversion symmetry. If we mirror the entire system except for the pump field, including both the dielectric structure and the optical signal fields, about the center plane of the metasurface/nanoantenna we find the system unchanged apart from a sign flip of the signal helicity. As the local electric field rotates in opposite directions for opposite helicities, right handed CPL will experience the same nonreciprocal response as left handed CPL but with the roles of forward and reverse direction exchanged, which we confirm in the supporting information. Just as cross polarizers are needed to turn a magneto-optic crystal into a Faraday isolator, a single handedness of CPL must be selected to fully suppress transmission through our metasurface in the reverse direction. A traditional way of building such a circular polarizer is given in the supporting information. From a nanophotonics perspective, thankfully, unlike time reversal symmetry, spatial inversion symmetry can be broken by structuring standard linear dielectric materials. Many examples of chiral metamaterials and metasurfaces, using both dielectric and metallic resonators, have appeared in the literature. So, while beyond the scope of the present study, we believe that designing chiral versions of the devices shown in Figs. 2-4 which are capable of full isolation should be well within reach.^{60-62”}

60. Wang, Z., Cheng, F., Winsor, T. & Liu, Y. Optical chiral metamaterials: a review of the fundamentals, fabrication methods and applications. *Nanotechnology* **27**, 412001 (2016).
61. Zhu, A. Y. *et al.* Giant intrinsic chiro-optical activity in planar dielectric nanostructures. *Light Sci. Appl.* **7**, 17158 (2018).
62. Hu, J. *et al.* All-dielectric metasurface circular dichroism waveplate. *Sci. Rep.* **7**, 41893 (2017).

The following section has also been added to the SI:

Designing a circular polarizer

In the main text we focus on breaking time reversal symmetry. For circularly polarized light, helicity is conserved under time reversal. So, the time reverse of a left handed wave is still left handed but travelling in the opposite direction. Waves with opposite helicity are instead related by inversion symmetry. S5 shows the power dependent response of our inversion symmetric metasurface to right handed CPL. Strong nonreciprocity with almost identical power dependence is seen except for the exchange of forward and reverse directions.

S5: Directional transmittance as a function of Pump irradiance corresponding to metasurface considered in fig3 of the main text, but with right handed signal.

Waves with opposite helicity can be distinguished by arranging passive dielectric materials asymmetrically in space. S6 shows schematically how this can be achieved with a simple stack of traditional optical

S6: Circular polarizer constructed from linear polarizer sandwiched between two quarter waveplates. a) left incidence and b) right incidence.

components. The stack consists of a linear polarizer sandwiched between two quarter waveplates. By rotating one quarter waveplate by 90 degrees with respect to the other, the incident helicity will be preserved upon passive through. But in-between the waveplates left and right handed incident waves are transformed into orthogonal linear polarisations. The linear polarizer is then able to select a particular handedness to be transmitted while the other is reflected. Comparing S6a and S6b, we can see that Left handed light is transmitted in both directions while right handed light is reflected with helicity preserved. Placing our nonreciprocal spin selective metasurface in front of this device would produce full optical isolation. This device could also be shrunk down to the nanoscale using resonant chiral nanoantennas, or even incorporated into our nonreciprocal scheme by inducing stimulated Raman scattering in a chiral metasurface.

1. The presence of the rotating phonon mode seems to be essential for realizing the SRS-based optical isolator, but it is not conveyed clearly in the text (I guess it is a special case of equation 4).

We thank the reviewer for highlighting that this important point was not clear. For clarification, figure 1 and the discussion of equation 4 has been updated:

“Nonreciprocal light flow in a uniform medium is dependent on the difference between the off-diagonal elements of the susceptibility, $|\chi_{yx} - \chi_{xy}|$. For SRS, we can see that $|\chi_{yx} - \chi_{xy}| \propto (ac - b^2)Im(p_x p_y^*)$, and so nonreciprocity will result if the pump polarization possesses some ellipticity, $Im(p_x p_y^*) \neq 0$, and, $ac \neq b^2$. By a 45 degree coordinate rotation, $ac = b^2$ can be seen to represent a phonon which couples exclusively to electric fields along a single cartesian axis; this condition is only possible for extremely anisotropic crystals. Therefore, beyond previous observations of time reversal symmetry breaking in silicon waveguides^{31,36}, we expect Raman-based nonreciprocity to be ubiquitous in a range of classical and quantum photonic materials and devices.

Equation 4 also reveals an interesting special case which will be the focus of the rest of this paper. If $a = c$ and $b = 0$ or $a = c = 0$ and $b \neq 0$, a circularly polarized pump, $p_x/p_y = \pm i$, produces an SRS susceptibility proportional to $(1, \mp i; \pm i, 1)$. Examples of specific phonons that support these kinds of Raman scattering are shown in Fig. 1c. The left panel shows the z polarized phonon in silicon which possesses S_4 symmetry about z and so mixes x and y polarized photons. The right

panel shows the z polarized phonon of blue phosphorene³⁷ which exhibits an isotropic response and thus a diagonal Raman tensor. The key feature of the phonons depicted in Fig. 1c is that they couple equally to x and y polarized components of the incident electric field. The rotation, or spin, of the pump electric field can therefore be imprinted on to the host crystal. Many other materials support spin selective Raman tensors, including, Diamond, GaAs, GaP, Barium Nitrate, and Potassium Gadolinium Tungstate.

Fig. 1: Nonreciprocal stimulated Raman scattering. a) Energy diagram for Raman decay of a pump photon into a Stokes photon and a phonon b) Nonreciprocal polarization rotation due to Faraday effect in magnetized medium. c) Phonons that produce spin polarized nonreciprocal SRS for CPL propagating in z. Mulliken symbols and x-y Raman tensors are provided for silicon (left) and blue phosphorene (right) d) One-way amplification due to stimulated Raman scattering in silicon pumped with circularly polarized light.

2. It is unclear if the design actually prohibits transmission of the reverse signal, or alternatively, if the design allows transmission of the reverse signal but does not provide gain for it. Perhaps this could be better emphasized if already stated.

For clarity, the discussion of figure 3 has been modified as follows:

“Fig. 3c compares a resonant signal passing through the metasurface in forward and reverse propagation directions for increasing pump power. As seen by the persistently low value of the blue curve in Fig. 3c, our metasurface is fully opaque under reverse illumination, but becomes transparent for forward illumination when pumping above $7\text{mW}/\mu\text{m}^2$, peaking at $27\text{mW}/\mu\text{m}^2$; higher pump powers reduce the nonreciprocal response due to the onset of lasing. Note that this result is distinct from the case illustrated in Fig. 1d for a bulk silicon crystal; there, the signal incident from one direction is amplified while from the opposite direction transmission is neither amplified nor suppressed. Our ultrathin silicon metasurface, with a thickness of just $\lambda_s/7$, therefore exhibits highly nonreciprocal transmission.”

3. It could be helpful to better define some terms or clarify meaning, especially for a more general audience:

a. “undepleted pump approximation”

“Physically this approximation is equivalent to ignoring the negligible absorption of the pump wave by the signal wave.” has been added after this phrase for clarity.

b. “fixed rotation of the bias field phonon system”

The sentence containing this phrase has been changed to: “Importantly, the directionality of this process results purely from the modified effective material properties which appear from the point of view of the Stokes wave as a continuously rotating phonon being driven by the rotating electric pump field, which holds right down to the atomic scale.” For clarity.

c. “traceless Raman tensor”

We have replaced “traceless Raman tensor” in the original text with a concrete example for Si. For this diamond cubic lattice, the diagonal elements are identically zero. The new text states:

“For a concrete demonstration we choose silicon which has a rather strong Raman transition, $\chi_{res} = 11.2 \times 10^{-18}(\text{m}/\text{V})^2$, $\Omega = 15.6\text{THz}$ and $\Gamma \approx 53\text{GHz}$, as well as a symmetric Raman tensor with zeroes on the diagonal when working in the coordinate frame of its diamond cubic lattice³⁸”

Reviewers' Comments:

Reviewer #1:

Remarks to the Author:

I have read the revised version of this manuscript, and the authors' responses to the comments by the two referees.

From my perspective, the authors have adequately/satisfactorily addressed in their replies all the comments I previously made, and have accordingly amended, improved and further clarified the main paper and its supplementary information.

Thus, in light also of the recent, wider and substantial, interest in efficient (and ideally nanoscale) nonreciprocity schemes, as alluded to in my previous report, I feel that this work makes an interesting, novel and timely contribution to the broader field, which is now ready to appear in the journal as is.

Reviewer #2:

Remarks to the Author:

This manuscript was already of very high quality at the time of my initial review. I raised no major scientific concerns, and it appears that was also true for the other referee. My concerns were minor, and I believe the authors have adequately addressed my concerns, as well as the issues identified by the other referee. I believe the manuscript would now be ready for publication.